# Choriocapillaris Flow Deficits Quantification in Hydroxychloroquine Retinopathy Using Swept-Source Optical Coherence Tomography Angiography

**DOI:** 10.3390/jpm12091445

**Published:** 2022-09-01

**Authors:** Safa Halouani, Hoang Mai Le, Salomon Yves Cohen, Narimane Terkmane, Nabil Herda, Eric H. Souied, Alexandra Miere

**Affiliations:** 1Department of Ophthalmology, Centre Hospitalier Intercommunal de Créteil 40, Avenue de Verdun, 94100 Créteil, France; 2Department of Ophthalmology, Hôpital Henri-Mondor AP-HP, 94100 Créteil, France

**Keywords:** hydroxychloroquine, toxicity, choriocapillaris, oct angiography

## Abstract

This study aims to quantitatively analyze choriocapillaris (CC) alterations using swept-source optical coherence tomography angiography (SS-OCTA) in eyes presenting with hydroxychloroquine (HCQ) toxic retinopathy and to compare it to patients under HCQ without toxic retinopathy and to healthy controls. For image analysis, CC en-face slabs were extracted from macular 6 × 6 mm SS-OCTA scans and a compensation method followed by the Phansalkar local thresholding was performed. Percentage of flow deficits (FD%) and other related biomarkers were computed for comparison. Fourteen eyes (7 patients) presenting with HCQ toxic retinopathy, sixty-two eyes (31 patients) under HCQ without signs of toxicity, and sixty eyes of 34 healthy controls were included. With regards to FD%, FD average size, and FD number there was a significant difference between the three groups (*p* < 0.05 with radius 4 and radius 8 pixels). Eyes presenting with HCQ toxicity had significantly higher FD% and average size, and a significantly lower number of FDs, with both radius 4 and 8 pixels. In conclusion, FD quantification demonstrates that CC involvement is present in HCQ toxic retinopathy, therefore giving pathophysiological insights with regards to the CC as being either the primary or secondary target of HCQ toxicity.

## 1. Introduction

Hydroxychloroquine (HCQ) is recognized for its anti-inflammatory, anti-thrombotic, and recently identified immunomodulatory properties [1,2]. Its indications in pediatric inflammatory disorders, neurology, and oncology predict a large and growing cohort of long-term users [2,3,4]. In addition, during the covid19 pandemic, HCQ and chloroquine have been employed in the clinical setting to manage SARS-CoV-2 infection [5].

Side effects include ophthalmological complications, namely corneal verticillate pigment deposition, accommodation disorders, and potential retinal dysfunction. The retinal toxicity was initially considered rare (estimated occurrence in 0.5–2.0% of long-term users) as the diagnosis was based on late-stage bull’s eye maculopathy [6].

In the absence of early diagnosis and without stopping the treatment, the toxic retinopathy can lead to severe bilateral visual loss, causing severe visual handicap. The importance of monitoring has been highlighted in the American Academy of Ophthalmology [7] and the Royal College of Ophthalmology [8] recommendations and, more recently, the American College of Rheumatology, American Academy of Dermatology, Rheumatologic Dermatology Society, and American Academy of Ophthalmology 2020 Joint Statement [9]. Modern screening tools allow the detection of early-stage toxicity. At least one objective structural test and one subjective functional test should be performed on these patients [7]. Since automated visual fields and spectral-domain optical coherence tomography (OCT) are widely available, these are the recommended tests for primary routine screening [7,8].

Based on spectral-domain OCT analysis, previous studies have proved that HCQ toxicity resulted in significant outer retinal layer volumetric thinning, affecting the photoreceptor layer, the interdigitation zone, and the ellipsoid zone (EZ) from the early stages of retinal involvement [10,11]. In addition, more severe impairment, consisting of retinal pigment epithelium (RPE) damage and choroidal thinning, may also occur in the late stages of HCQ toxicity [11,12]. Moreover, recent literature suggests that, besides the well-described (outer) and inner retinal involvement, choroidal thinning also plays a role in the physiopathology of HCQ toxicity [13,14].

Of recent, optical coherence tomography angiography has been suggested as a possible screening tool for HCQ-induced retinal alterations [14,15]. However, most of the studies emphasized the differences in macular vascular density, at the level of the capillary plexus or with regards to the foveal avascular zone, between HCQ retinal toxicity eyes and healthy controls [15,16,17]. Therefore, this study aims to quantify and compare the choriocapillaris (CC) alterations between eyes with HCQ retinopathy, eyes of patients treated by HCQ without retinal toxicity, and healthy controls.

## 2. Materials and Methods

### 2.1. Study Participants

In this cross-sectional cohort study, we prospectively included consecutive patients from the Department of Ophthalmology of University Paris Est, in Creteil, France, and from the Department of Ophthalmology of Henri Mondor University Hospital from January 2021 to June 2021. The study was conducted in agreement with the Declaration of Helsinki for research involving human subjects and was approved by the Ethics Committee. In addition, approval from the Institutional Review Board (IRB) was obtained prior to this study.

Inclusion criteria were: (1) previous treatment with HCQ for at least one year; (2) clear ocular media to ensure good image quality. In addition, a sex-matched, age-matched control group was included. Hence, two groups were distinguished: the HCQ group, called group 1, and the healthy sex- and age-matched control group (group 2). Furthermore, within group 1, two subgroups were identified according to the presence/absence of toxicity: (a) no toxicity (group 1a) and (b) toxic retinopathy (group 1b).

For group 1, the disease for which the treatment was prescribed, the daily dose, the duration of treatment, the body weight, and the cumulative dose in years were recorded.

According to Marmor’s definition and grading of toxicity [6,18], diagnostic criteria for HCQ retinopathy were based on the presence of parafoveal or bull’s-eye retinal damage confirmed by at least three different tests, including fundus examination, macular visual fields, multifocal electroretinogram (mfERG), spectral-domain OCT (SD-OCT), and Fundus autofluorescence (FAF).

Patients in group 1a did not show any abnormalities in the macular field or objective testing, whereas those in group 1b had either bull’s-eye damage on fundus examination or FAF with RPE involvement on SD-OCT (visible retinopathy). Figure 1 shows an example of multimodal imaging of a patient with toxic retinopathy, included group 1b.

The exclusion criteria were: (1) poor-quality SS-OCTA images with a signal strength index lower than 7, as recommended by the manufacturer; (2) significant motion artifact or incorrect segmentation; (3) other concomitant macular diseases in the study eye.

All subjects in group 1 underwent a complete ophthalmologic examination, including Snellen Best Corrected Visual Acuity (BCVA), which was converted to Logarithm of the Minimum Angle of Resolution (LogMAR) for statistical analysis, and slit lamp biomicroscopy. In addition, all patients underwent multimodal imaging, including infrared reflectance (IR), FAF, Macular Visual Field, mfERG, structural spectral-domain OCT (Spectralis HRA + OCT system, Heidelberg Engineering, Heidelberg, Germany), and SS-OCTA (Plex Elite 9000, Carl Zeiss Meditec, Inc., Dublin, CA, USA).

Subjects included in group 2 were healthy controls with no abnormalities on ophthalmological examination and no previous/current ophthalmological and systemic disease, having undergone BCVA testing and multimodal imaging, including SS-OCTA, as part of the present study.

### 2.2. Imaging Analysis

SS-OCTA imaging was performed using Plex Elite 9000. The macula was imaged using a 6 × 6 mm scan centered on the fovea. The study did not include images with poor signal strength (signal strength index <7). CC was analyzed using a custom 15 μm thick slab starting 16 μm below the RPE/Bruch Membrane, as previously described [19]. The en-face flow and structural images from this custom CC segmentation were then imported into Fiji Software (National Institute of Mental Health, Bethesda, MD, USA). As per recent literature [19,20,21,22], a compensation algorithm is used for CC analysis, using both the en-face flow and structural images to account for signal attenuation. In detail, an inverse transformation using the Fiji “Invert” function was applied to the en-face CC structure image [21]. Then, a Gaussian blur filter was applied for image smoothing. After multiplication between the en-face CC flow image and the processed en-face CC structure image using “Image Calculator”, a compensated en-face CC image was obtained. Finally, the Phansalkar local thresholding method using a window radius of both 4 pixels and 8 pixels was performed for binarization of the compensated en-face CC flow image to obtain a quantitative analysis of the flow deficits (FD). Figure 2 summarizes the image processing method.

The main outcome measures are the percentage of FD (FD%), the number and size of FD, and the total FD area. The number of FDs was the number of contiguous black pixels representing the FDs. The size of FDs was the mean area in μm^2^ of the contiguous black pixel area. Total FDs area represents the area in μm^2^ of all the FDs. FD% was computed as the total FDs area on the total image (black and white pixels).

### 2.3. Statistical Analysis

Values of quantitative variables were reported as means ± standard deviations, whereas values of categorical variables were expressed as counts and percentages.

The results from groups 1a, 1b, and 2 were analyzed and compared. In addition, parametric student’s T-test, and nonparametric methods (Mann-Whitney tests, Kruskal Wallis) were conducted to assess differences in quantitative variables. In all analyses, *p*-values < 0.05 were considered statistically significant.

The relationship between the CC (considered dependent variables) and the age, treatment duration, daily dose in milligrams per kilogram, and cumulative dose was investigated using multiple linear regression analysis.

## 3. Results

### 3.1. Patient Demographics and Main Clinical Features

Seventy-six eyes of 38 patients undergoing HCQ treatment (group 1) and sixty eyes of 34 healthy controls (group 2) were included in this study. Among the eyes included in group 1, sixty-two eyes (31 patients) did not have any signs of HCQ toxicity (subgroup 1a), while fourteen eyes (7 patients) presented with HCQ toxicity (subgroup 1b).

Within group 1, most patients (76.31%) were female, of which 80.65% were in group 1a, and 57.15% were in group 1b. The mean age of patients in group 1 was 49.89 ± 11.68 years versus 54.30 ± 14.84 in group 2. The mean age in the toxicity subgroup (1b) was 57.71 ± 3.49 versus 48.48 ± 12.18 years in the no toxicity subgroup (1a). There was no significant difference regarding age or gender between groups 1a, 1b, and 2 (*p*-value = 0.2566 and 0.993, respectively).

For group 1a, the mean treatment duration was 9.90 ± 5.38 years, while for group 1b, the treatment duration was 15.85 ± 9.04 years. The mean daily dose in milligrams per kilogram of weight was 5.75 ± 0.77 among the patients presenting with toxic retinopathy (1b), while it was 5.56 ± 1.09 in patients without toxicity (1a).

The main diseases for which HCQ was prescribed were systemic lupus erythematosus (SLE) in 46.30% of cases, polyarthritis rheumatoid in 9.7%, and Gougerot Sjogren syndrome in 4.8%.

There was no significant difference in BCVA between the three groups. However, BCVA was significantly lower among patients in group 1b than in group 1a (*p* value = 0.008). All patient demographics and clinical data are summarized in Table 1.

### 3.2. Choriocapillaris Analysis

#### 3.2.1. Choriocapillaris Analysis Using the Phansalkar Local Thresholding Method with a Window Radius 4 of Pixel

The difference between the three groups (1a, 1b and 2) was significant for FD%, FD average size, FD number, and FD total area (*p*-value 0.00137, 0.00186, 0.00194, 0.00137) (Table 2).

Compared to group 2 and group 1a, eyes presenting with toxic retinopathy included in groups 1b had a significantly higher percentage of FD (FD% 48.96 ± 7.63, *p*-value respectively 0.0092 and 0.00064) and significantly lower number of FD (mean count 5221.14 ± 4123.72, *p*-value 0.0053 and 0.00078) (Table 3).

The average size of FD was 9415.084 µm^2^ ± 14,453.98 and was significantly larger in group 1b compared to group 1a (*p*-value 0.00078).

The total area was significantly higher in toxicity group 1b (17.62 ± 2.74 mm^2^) compared to group 2. (*p*-value 0.0092).

There was no significant difference in terms of average size, total area, number, and percentage of FD between the control group (group 2) and the no-toxicity group (group 1a) (*p*-value respectively 0.515, 0.132, 0.183, and 0.1324).

Figure 3 shows an example of CC analysis using the Phansalkar local thresholding method in a patient’s eye from group 1b (toxic retinopathy).

#### 3.2.2. Choriocapillaris Analysis Using the Phansalkar Local Thresholding Method with a Window Radius of 8 Pixels

The difference between the three groups was significant in the FD%, the average size of FDs, the number of FDs, and FD total area (*p*-value 0.001845, 0.001520, 0.001845, 0.00161) (Table 2).

Compared to the control group and the no-toxicity group, respectively, eyes presenting with toxicity had a significantly higher percentage of FD (mean FD% 49.41 ± 7.42, *p*-value respectively 0.009453 and 0.00061) and significantly lower number of FD (mean count 4719.85 ± 3405.69, *p*-value 0.0050 and 0.00068) (Table 3).

The average size of FD was 8279.539 µm^2^ ± 10091.572 and was significantly larger in the toxicity group compared to the no-toxicity group (*p*-value 0.00064).

The total area was significantly higher in the toxicity group (17.78 ± 2.67 mm^2^) compared to the controls and the no-toxicity group (*p*-value = 0.0052 and 0.00061).

Figure 4 shows an example of CC analysis using the Phansalkar local thresholding method in a control’s eye (group 2).

The difference between group 1a (no-toxicity group) and group 2 (healthy controls) was not significant for FD%, FD average size, FD number, and FD total area (*p*-value respectively 0.124, 0.392, 0.173, and 0.389).

There was no significant association in multiple linear regression between age, treatment duration, daily dose in milligrams per kilogram, cumulative dose and flow deficit counts, percentage, total area, and average area both in window radius of 8 and 4 pixels.

## 4. Discussion

In this cross-sectional study, we prospectively investigated the quantitative CC changes in patients under HCQ with or without retinal toxicity and compared these choroidal changes to the healthy control group. In order to quantify the decrease in detected flow in the CC, we measured the percentage of FD, the number of FD, the total area, and the average area of FDs using both window radii of 8 and 4 pixels. All these CC variables were significantly different between the three groups. Eyes presenting with HCQ retinopathy (group 1b) were found to have a significantly higher percentage of FD but also a larger total area and a lower number of FDs than eyes under HCQ but without HCQ toxicity (group 1a) and age- and gender-matched healthy controls. This raises a crucial pathophysiological question on whether the CC impairment found in severe stages of HCQ toxicity is secondary to the loss of photoreceptors and RPE or if the loss of CC itself is secondary to the toxic effect of HCQ and therefore precipitates photoreceptor and RPE loss. In our series, the group treated with HCQ but without HCQ toxicity (group 1a) did not have significantly more FDs than the control group would support the former hypothesis, but there are several variables to consider. Among these variables, OCTA CC image processing plays a central role.

Following recent recommendations [19,21,22,23] on CC flow quantification and in the light of CC planar geometry, we explored the non-perfused area of the CC (i.e., FDs) using a compensation algorithm. In addition, CC quantification is highly variable, depending on the type of instrument (spectral domain/swept source) [21,23], CC slab selection, thresholding method, and compensation method. This quantification has investigated CC impairment in different retinal and choroidal diseases, such as advanced and intermediate age-related macular degeneration and macular neovascularization [20,24,25]. Moreover, the CC analysis in our study was performed on images from a swept-source OCTA instrument which operates at a wavelength of 1050 nm, as opposed to the spectral domain OCTA, which uses a shorter wavelength of 840 nm and thus has the inconvenient of limiting light penetration into the choroid [26,27,28]. We also opted for manual slab selection. Although automated segmentation works correctly for segmenting the RPE-Bruch’s membrane limit in healthy eyes, previous reports have proved that it is less accurate in diseased eyes due to the erratic automatic identification of the RPE [20,23]. Chu et al. previously used the CC slab in our study [20]. It was defined as a 15 mm thick slab starting 16 mm under the RPE/Bruch’s membrane and was adjusted manually in cases of segmentation errors. As for the thresholding method, most studies investigating the CC have adopted a local threshold (Phansalkar method) that uses a small window rather than the entire image to define the threshold for binarizing each pixel. Chu et al. [20] have recently investigated the most accurate use of this threshold for CC binarization. In the present study, we performed the analysis using both window radii, as even the slight differences in slab selection and image processing methods may lead to an important variation in the CC quantitative variables. As described by Zhang et al. [23], we used the compensation strategy to overcome the shadowing effect induced by the RPE-Bruch’s membrane complex on the CC. Ledesma-Gil G et al. [29] have reported that imaging compensation may also cause undesired artifacts. Shi Y et al. [30] suggested that compensated and non-compensated (i.e., original) en-face OCTA images of the CC may be binarized to isolate pixels representing areas of flow signal from those constituting signal deficits (i.e., FD). Hence, by using a standardized CC image processing method, we are reassured of the reliability of our results.

Regarding previous OCTA studies, most reports focused on the analysis of vessel density (VD) in superficial and deep retinal plexuses in eyes without signs of toxicity. In their prospective cross-sectional study, Goker et al. [16] compared both VD of macular capillary plexuses and foveal avascular zone (FAZ) areas of patients receiving HCQ for more than five years with those of age- and sex-matched controls. Foveal VD at the superficial capillary plexus (SCP) and deep capillary plexus (DCP) level was significantly lower in the HCQ group than in the control group. Forte et al. [17] evaluated swept-source OCTA outcomes in twenty eyes of ten patients treated with HCQ for more than five years. The VD in all retinal capillary plexuses was automatically generated. The VD in all capillary plexuses was reduced, and the FAZ area was larger than controls [17]. In the study by Bulut et al. [15], patients in a high-risk group (treatment duration of more than five years) had significant vascular density loss. In addition, the high-risk group decreased automatically generated retinal and choroidal perfusion density markedly [15].

Mihailovic et al. [31] analyzed all four layers in spectral domain OCTA only in patients with no signs of HCQ retinopathy. Images were converted into greyscales, and each pixel was attributed to a value representing the strength of the decorrelation signal. VD in the CC was calculated as the mean decorrelation value of all pixels in the images (arbitrary unit, AU) [31]. The CC analysis showed a reduced VD in SLE patients compared with healthy controls [31]. Other authors [14,17,32] have qualitatively reported CC signal voids in the areas of the RPE defect on OCTA, but none of the above have reported FDs quantification. For instance, following a qualitative CC analysis, Ahn et al. [32] found that in patients with severe retinopathy, the choroidal involvement continued to progress even after drug cessation [32].

Most studies on HCQ retinal toxicity confirm that the damage caused by HCQ starts at the outer retina [11,33] and that the inner retina is relatively intact in eyes with HCQ retinopathy [11]. However, the (inner) choroid has not been addressed quantitatively as a damage site of HCQ. Nevertheless, as stated above, the fact that the CC impairment is present in HCQ toxic retinopathy but not in the group treated by HCQ without HCQ toxicity supports the hypothesis that the alteration of the CC may be secondary to that of the outer retina.

The pathogenesis of HCQ retinal toxicity is still unclear. The primary site of drug toxicity is known to be the photoreceptor layers, with possible secondary involvement of the RPE [6,34]. Choroidal alterations are reported in the pathogenesis of several retinal and choroidal diseases [35,36] as CC feeds the outer retinal layers, including the RPE and photoreceptor layers. The CC, the RPE, Bruch membrane, and photoreceptors form a symbiotic unit indeed, and the thinning of the outer nuclear layer may also reflect the damage of this unit, ultimately resulting in large regions of RPE atrophy and choroid alterations [37].

Our study has several limitations, including the small sample size of the toxic retinopathy group, its cross-sectional design, and the inclusion of each patient’s eyes. Following prospective longitudinal reports with a larger sample size investigating the CC impairment in eyes with HCQ toxicity may allow a better understanding of disease physiopathology and thus enhance the screening protocols.

In conclusion, this study reports on the CC changes occurring in HCQ retinal toxicity, showing that eyes with HCQ toxic retinopathy had a significantly greater amount of CC hypoperfusion (and thus a higher percentage of FD). In addition, our results expand the current knowledge of the choroidal involvement in the pathogenesis of HCQ retinopathy, especially in the late stages. Lastly, the CC perfusion, if replicated in future studies, might be considered a valuable marker for monitoring, and assessing subjects taking this treatment.

## Figures and Tables

**Figure 1 jpm-12-01445-f001:**
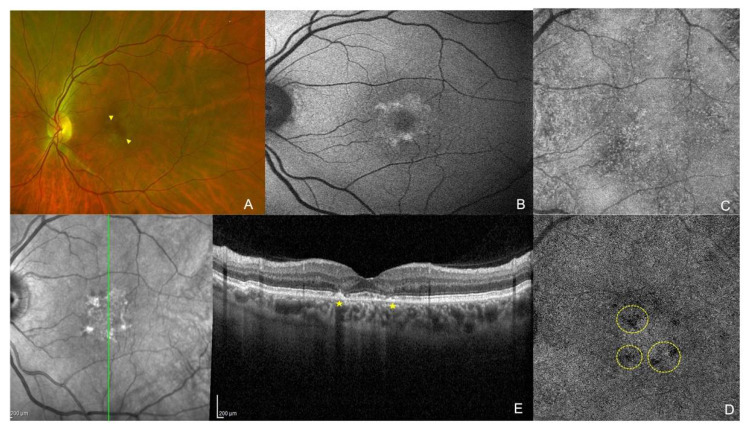
Multimodal imaging of a left eye of a patient presenting with hydroxychloroquine retinopathy: (**A**) color retinal photography revealing yellowish annular lesions in the foveal area (yellow arrowheads); (**B**) Fundus autofluorescence showing macular hyperautofluorescent ring delimitating the macular lesions; (**C**) choriocapillaris en-face structural image in swept-source optical coherence tomography (SS-OCT) angiography and en-face flow image; (**D**) showing flow deficits (yellow circles); (**E**) Spectral Domain OCT vertical B scan showing parafoveal outer retinal layers and ellipsoid disruption defining the typical flying saucer sign (yellow stars).

**Figure 2 jpm-12-01445-f002:**
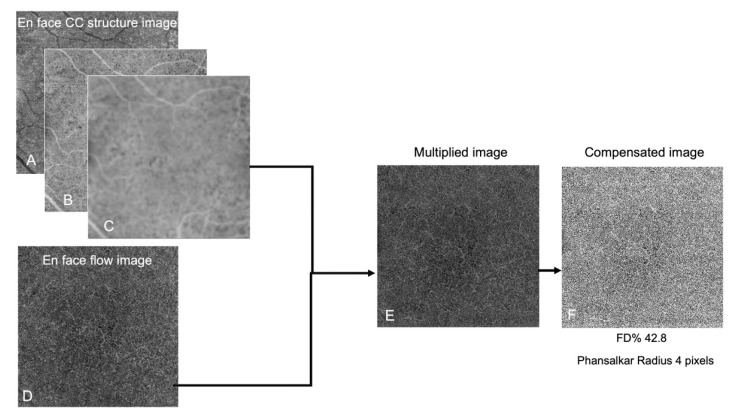
The image processing method used to investigate the choriocapillaris flow. En-face choriocapillaris (CC) structure image (**A**) underwent an inverse transformation using the Fiji “Invert” function (**B**). Next, smoothing was obtained using a Gaussian blur filter (**C**). Next, multiplication between the en-face CC flow image (**D**) and the processed en-face CC structure image (**C**) was performed using the “Image Calculator” (**E**). After multiplication between the en-face CC flow image and the processed en-face CC structure, a compensated en-face CC image was generated by Fiji (**F**). In this example, the compensated en-face CC flow image binarization was performed using a Phansalkar Radius of 4 pixels. The flow deficit percentage (FD%) was 42.82%.

**Figure 3 jpm-12-01445-f003:**
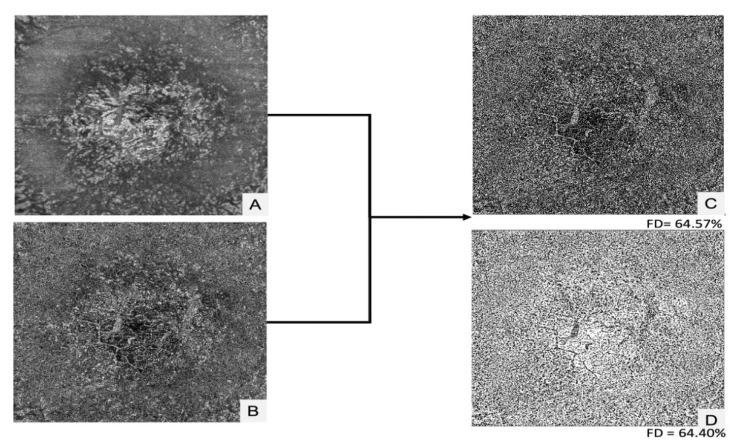
Choriocapillaris analysis using the Phansalkar local thresholding method in a patient’s eye with hydroxychloroquine retinal toxicity: (**A**) Choriocapillaris structure en-face image; (**B**) Choriocapillaris Flow en-face image; (**C**) Resulting image using the Phansalkar’s local thresholding method with a window radius of 4 pixels; (**D**) Resulting image using the Phansalkar’s local thresholding method with a window radius of 8 pixels.

**Figure 4 jpm-12-01445-f004:**
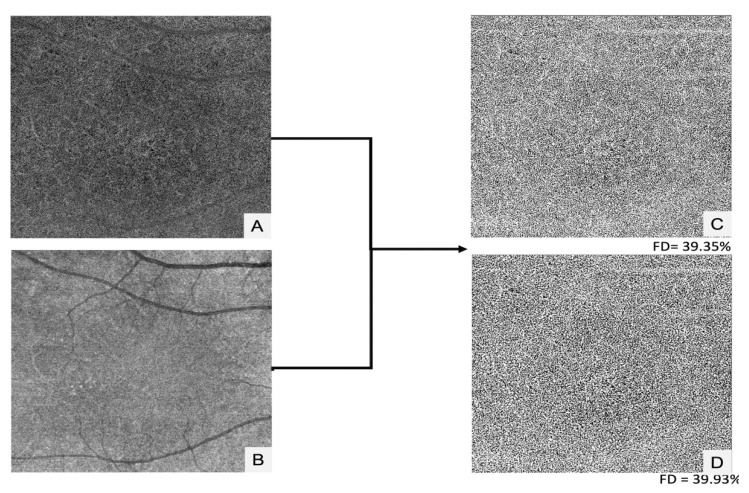
Choriocapillaris analysis using the Phansalkar local thresholding method in a healthy control’s eye: (**A**) Choriocapillaris structure en-face image; (**B**) Choriocapillaris Flow en-face image; (**C**) Resulting image using the Phansalkar’s local thresholding method with a window radius of 4 pixels; (**D**) Resulting image using the Phansalkar’s local thresholding method with a window radius of 8 pixels.

**Table 1 jpm-12-01445-t001:** Main clinical and demographic features of the study group. Group 1a: eyes of patients under HCQ without retinal toxicity; Group 1b: eyes with toxic HCQ retinopathy; Group 2: eyes of healthy controls; * Kruskal Wallis, ** chi2 test, *** *t*-test; NA: non-applicable.

	Group 2 (Mean ± SD) or *n* (%)	Group 1a (Mean ± SD) or *n* (%)	Group 1b (Mean ± SD) or *n* (%)	*p*-Value
Age	54.30 ± 14.84	48.483 ± 12.18	57.71 ± 3.49	0.2566 *
Gender (males)	23.5	19.35	42.85	0.993 **
BCVA (LogMAR)	0.0194 ± 0.0398	0.0096 ± 0.029	0.128 ± 0.1326	2.29 *
Treatment duration (years)	NA	9.90 ± 5.38	15.85 ± 9.04	0.137 ***
Daily dose (mg/Kg)	NA	5.56 ± 1.09	5.75 ± 0.77	0.59 ***
Cumulative dose (grams)	NA	1862.67 ± 592.72	2294.28 ± 1274.63	0.11 ***

**Table 2 jpm-12-01445-t002:** SS-OCTA analysis of choriocapillaris flow on the 6 × 6 image: comparisons between the three-group using radius 4 and 8. FD: flow deficit; Group 1a: eyes of patients under HCQ without retinal toxicity; Group 1b: eyes with HCQ toxic retinopathy; Group 2: eyes of healthy controls; * Kruskal-Wallis test.

		Group 1a	Group 1b	Group 2	Significance (*p*-Value) *
**Radius 4**	Percentage of FD%	36.76 ± 1.42	48.96 ± 7.63	45.61 ± 3.98	0.0013
	Average size of FD (µm^2^)	975.13 ± 117.83	9415.08 ± 14,453.98	3321.92 ± 1942.51	0.0018
	Total area of FD (mm^2^)	14.70 ± 1.29	17.62 ± 2.74	16.42 ± 1.43	0.0013
	Number of FD	13,639.5 ± 1120.76	5221.14 ± 4123.72	7085.5 ± 2965.60	0.0019
**Radius 8**	Percentage of FD%	41.47 ± 1.40	49.41 ± 7.42	46.07 ± 1.40	0.0018
	Average size of FD (µm^2^)	1157.83 ± 140.24	8279.53 ± 10,091.57	3551.65 ± 1895.34	0.0015
	Total area of FD (mm^2^)	13.50 ± 0.50	17.78 ± 2.67	16.58 ± 1.39	0.0016
	Number of FD	11721 ± 982.87	4719.85 ± 3405.69	5324 ± 2449.41	0.0018

**Table 3 jpm-12-01445-t003:** SS-OCTA analysis of choriocapillaris flow: comparisons between eyes with retinal toxicity versus eyes under HCQ without toxicity and healthy controls. R4: analysis using the Phansalkar local thresholding method with a window radius of 4 pixels; R8: analysis using the Phansalkar local thresholding method with a window radius of 8 pixels; Group 1a: eyes of patients under HCQ without retinal toxicity; Group 1b: eyes with HCQ toxic retinopathy; Group 2: eyes of healthy controls; * *t*-test; ** Wilcoxon test.

	Group 1b	Group 1a	Group 2	Significance Group 1b vs. 2	Significance Group 1b vs. 1a
Percentage of flow deficits (FD%) R4	48.96 ± 7.63	36.76 ± 1.42	45.61 ± 3.98	0.0092 *	0.00064 **
Total Area of flow deficits R4 (mm^2^)	17.62 ± 2.74	14.70 ± 1.29	16.42 ± 1.43	0.0092 *	0.00064 **
Average size of flow deficits R4 (μm^2^)	9415.08 ± 14,453.98	975.13 ± 117.83	3321.92 ± 1942.51	0.10 *	0.00078 **
Number of flow deficits R4	5221.14 ± 4123.72	13,639.5 ± 1120.76	7085.5 ± 2965.60	0.0053 *	0.00078 **
Percentage of flow deficits (FD%) R8	49.41 ± 7.42	41.47 ± 1.40	46.07 ± 1.40	0.0094 *	0.00061 **
Total Area of flow deficits R8 (mm^2^)	17.78 ± 2.67	13.50 ± 0.50	16.58 ± 1.39	0.0052 *	0.00061 **
Average size of flow deficits R8 (μm^2^)	8279.53 ± 10,091.57	1157.83 ± 140.24	3551.65 ± 1895.34	0.077 *	0.00064 **
Number of flow deficits R8	4719.85 ± 3405.69	11721 ± 982.87	5324 ± 2449.41	0.0050 *	0.00068 **

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
