# Peer review of "Choriocapillaris Flow Deficits Quantification in Hydroxychloroquine Retinopathy Using Swept-Source Optical Coherence Tomography Angiography"

_jpm, 2022, doi:10.3390/jpm12091445_

Round 1
Reviewer 1 Report
In this cross-sectional study, the authors quantitatively analyzed choriocapillaris (CC) alterations using SS-OCTA in eyes presenting with hydroxychloroquine (HCQ) toxic retinopathy and compared them to patients under HCQ without toxic retinopathy and to healthy subjects.
The manuscript is well organized and conclusions are supported by the results. The topic is appealing for the readers.
I recommend some minor revisions:
- A revision by a native English speaker should be performed.
-In the introduction, please cite the following article dealing with HCQ retinopathy (PMID 32946004)
- Line 111: Please modify with 'by using' instead of 'using'
- In the conclusion, the relatively small sample size may represent another limitation of the study. Other larger-scale studies should confirm these results. Please discuss it.
Author Response
Dear Editor,
We are grateful for the review of our manuscript entitled "Choriocapillaris flow deficits quantification in hydroxychloroquine retinopathy using swept-source optical coherence tomography angiography." The reviewers raised essential questions that we have addressed and incorporated into the revised version of the manuscript and the revised Figures and table. Please find attached the responses to the reviewers' comments.

Reviewer 2 Report
The authors present an interesting study on analysis of choriocapillary flow defects in HCQ retinopathy. The manuscript is well written.
However, one of the major drawbacks of this study is the small sample size in the toxicity sub-group.
Second, the focal RPE thickenings in HCQ toxicity could result in shadow artifacts in the CC layer (as seen in figure 1 OCT B-scan).
It would have been interesting to see if there were any differences between non-toxic group and normal controls. As it might help to detect preclinical toxicity. The authors should mention the post hoc p-values between these two groups.
Minor comments:
Table2:Kindly replace "Pourcentage" and "seize" with "percentage" and "size"
The labels (A, B, C, D) in figure 3 are not clear. Kindly change the color to white or place a white text box behind them.
Author Response
Dear Editor,
We are grateful for the review of our manuscript entitled " Choriocapillaris flow deficits quantification in hydroxychloroquine retinopathy using swept-source optical coherence tomography angiography. The reviewers raised important questions that we have addressed and incorporated into the revised version of the manuscript and the revised Figures and table. Please see the attachment for the point-by-point for reviewers' comments.
